# Coupling spectral imaging and laboratory analyses to digitally map sediment parameters and stratigraphic layers in Yeha, Ethiopia

Vincent Haburaj[1]*, Sarah Japp[2], Iris Gerlach[2], Philipp Hoelzmann[1], Brigitta Schütt[1]

1 Institute of Geographical Sciences, Freie Universität Berlin, Berlin, Germany, 2 Orient Department, Deutsches Archäologisches Institut, Berlin, Germany

* vincent.haburaj@fu-berlin.de

**Data Availability Statement:** All relevant data are within the manuscript and its Supporting Information files. S2 is available here: https://doi.

## Abstract

Quantitative analyses of soil and sediment samples are often used to complement stratigraphic interpretations in archaeological and geoscientific research. The outcome of such analyses often is confined to small parts of the examined profiles as only a limited number of samples can be extracted and processed. Recent laboratory studies show that such selectively measured soil and sediment characteristics can be spatially extrapolated using spectral image data, resulting in reliable maps of a variety of parameters. However, on-site usage of this method has not been examined. We therefore explore, whether image data (RGB data and visible and near infrared hyperspectral data), acquired under regular field-work conditions during an archaeological excavation, in combination with a sampling strategy that is close to common practice, can be used to produce maps of soil organic matter, hematite, calcite, several weathering indices and grain size characteristics throughout complex archaeological profiles. We examine two profiles from an archaeological trench in Yeha (Tigray, Ethiopia). Our findings show a promising performance of RGB data and its derivative CIELAB as well as hyperspectral data for the prediction of parameters via random forest regression. By including two individual profiles we are able to assess the accuracy and reproducibility of our results, and illustrate the advantages and drawbacks of a higher spectral resolution and the necessary additional effort during fieldwork. The produced maps of the parameters examined allow us to critically reflect on the stratigraphic interpretation and offer a more objective basis for layer delineation in general. Our study therefore promotes more transparent and reproducible documentation for often destructive archaeological fieldwork.

## Introduction

In the past decade a growing number of institutions and authors postulate more transparent and reproducible research approaches [e.g. 1; 2]. In this context, archaeologists began to

org/10.5281/zenodo.3906210. S3 is available here: https://doi.org/10.5281/zenodo.3906216.

**Funding:** This research was funded by the Cluster of Excellence EXC264 Topoi (The Formation and Transformation of Space and Knowledge in Ancient Civilizations, Research Area A; Deutsche Forschungsgemeinschaft (DFG) project number 39235742.

**Competing interests:** The authors have declared that no competing interests exist.

critically reflect on multiple aspects of data generation, analysis and interpretation [e.g. 3; 4]. Keeping in mind the often destructive character of archaeological excavations, transparency and reproducibility are highly relevant factors for archaeological fieldwork which involves on-site documentation and interpretation. One major part of this fieldwork is the delineation and characterisation of stratigraphic layers. While often accompanied by selective quantitative analyses of the samples extracted, the description and interpretation of the stratigraphic material remain in part influenced partly by the subjective perception of the respective researcher at work [cf. 5]. In the sense of Marwick et al. 2017 [4], stratigraphic interpretation should therefore be based on a more open method, reproducible independent of the researcher's experience. In the study at hand we present an approach that—in the long run—aims to promote a transparent, reproducible method of stratigraphic delineation and interpretation that is based on spectral recordings, i.e. physical information.

The often time-consuming quantitative analyses that traditionally support stratigraphic interpretation remain restricted to selective data based on the sampled materials and their laboratory analysis. By combining this data with digital image data, recent studies have successfully transferred selected soil and sediment properties, as derived from samples, to entire areas of soil and sediment cores and profiles: Steffens and Buddenbaum 2012 [6] and Hobley et al. 2018 [7], for example, were able to successfully map soil organic carbon (SOC) throughout soil and sediment profiles by analysing laboratory recorded hyperspectral data. These studies extend the subject of digital soil mapping (DSM) which until lately has been mainly concerned with the mapping of soil characteristics based on geodata like aerial or satellite image data [e.g. 8–11]. Accordingly, digital maps of chemical and physical parameters throughout soil and sediment profiles could act as an additional layer of information for archaeological excavations and the interpretation of the excavated material. However, the studies of Steffens and Buddenbaum 2012 [6] and Hobley et al. 2018 [7] were carried out under controlled laboratory conditions. Transferability of the proposed method into a less controlled environment has been examined by Zhang and Hartemink 2019 [12] and Haburaj et al. 2020 [13], who show that RGB imaging and multispectral imaging are also suitable tools to map stratigraphic layers under fieldwork conditions. Zhang and Hartemink 2019 [12] examined the extrapolation of SOC content, pH values, grain size composition and weathering indices based on the RGB image data of a soil profile that was captured on-site. The resulting parameter-maps were highly accurate, which may be the result of their experimental design: a large number of samples (n = 90) were analysed for calibration from a small section (1.0 x 0.9 m) of a high-contrast Alfisol. Building upon these results, our study aims to predict the sediment properties of complex archaeological profiles from RGB and hyperspectral image data that were captured under fieldwork conditions. Our experimental setup features two archaeological sections (Figs 1 and 2). The sampling strategy used for these sections follows common practice during excavations. The quality of the parameter-maps thus produced was assessed by comparison with stratigraphic delineation by archaeological experts (as depicted in Fig 2) as well as multiple statistical parameters. Our results show a promising performance of the proposed method for an extended stratigraphic analysis of archaeological excavations.

## Study area and archaeological background

The settlement of Yeha is located 35 km to the northeast of Aksum and spreads over the eastern footslopes of several volcanic rock mountains [14, 15]. It is surrounded by a number of smaller periodically discharging channels which unite east of the settlement and subsequently traverse a fertile plain (Fig 1a). Besides a huge ancient settlement, two prehistoric monumental buildings are currently known within the area of the modern rural settlement of Yeha: the

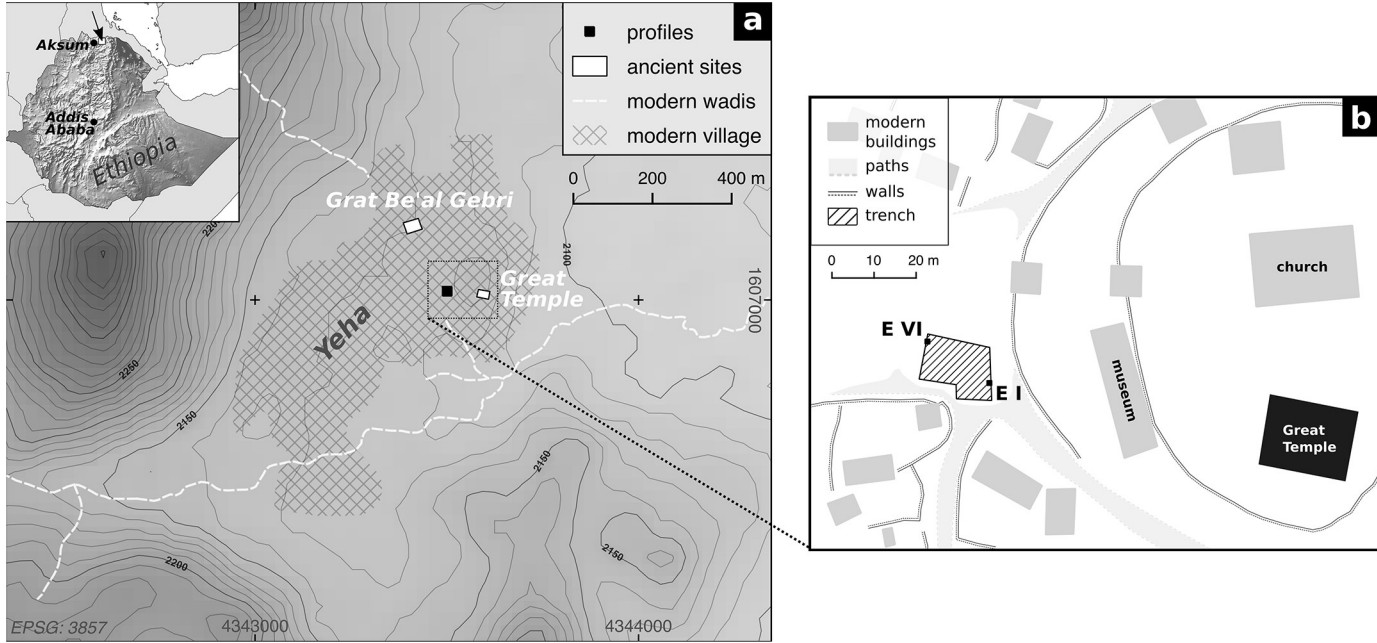

**Fig 1. Topographical map of the study area (a) and the archaeological profiles examined (b).** Elevation: SRTM data: SRTM data (1 arc-sec.; U.S. Geological Survey).

Great Temple in the east and the Grat Be'al Gebri in the north (Fig 1b). Both buildings are manifestations of a cultural transfer between indigenous societies located in todays northern Ethiopian highlands and the Sabaean society in South Arabia [16–20] in the first half of the first millennium BC. While the character of these interactions is still under debate [14, 16–25],

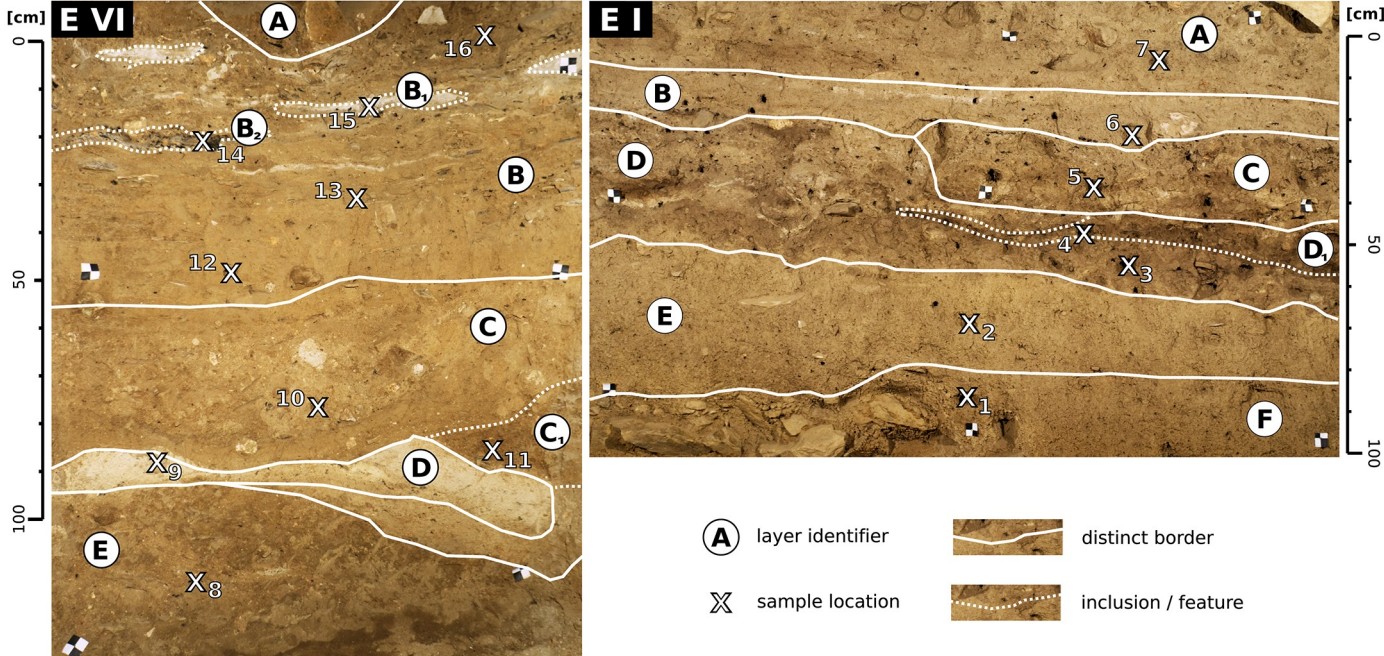

**Fig 2. Archaeological profiles EI and EVI.** The on-site stratigraphic interpretation is shown as white lines. Sediment samples were extracted at the 16 marked locations.

there are obvious similarities in both the architectural and archaeological remains such as the use of the Sabaean language and South Arabian script, monumental architecture and building decoration, Sabaean deities and cultic rites, and bronze and other elements of the material culture. The results of archaeological research in various parts of the modern settlement suggest that the site was populated at least from the late 2nd millennium BC onwards [26], likely by sedentary farmers [27]. From the early 1st millennium BC onwards Yeha developed as the center of the Ethio-Sabaean community until, according to present knowledge, the mid-1st millennium BCE [19, 20, 26].

The profiles examined in this study were recorded during excavations by the Ethiopian-German Archaeological Mission to Hawelti, Yeha and surroundings (conducted as a cooperation project between the Sanaa Branch of the Orient Department of the German Archaeological Institute [DAI], the Seminar for Oriental Studies of the Friedrich Schiller-University of Jena, the Authority for Research and Conservation of Cultural Heritage [ARCCH] and the Tigray Tourism and Culture Bureau [TCTB]) at Yeha in 2018 and 2019. They are located in the area of the modern settlement of Yeha (Tigray, Ethiopia).

### Profile description

Both profiles are part of an archaeological trench [26], found in the courtyard in front of the church compound; the Great Temple of Yeha is situated in the latter (Fig 1b). The examined areas of the profiles are shown in Fig 2. Profile EVI (western profile) covers several anthropogenic layers approximately 8 metres below the modern surface. The examined part of profile EI (eastern profile) involves several anthropogenic layers and a fine grained light brown horizontal layer which is present in many parts of the excavation but whose character is still under discussion. A detailed description of the two profiles EI and EVI is given in Table 1.

The two profiles do not overlap spatially and come from different depths of the archaeological trench. Additionally they date to different times: EI covers Aksumite and post-Aksumite times (1st half 1st to 1st half 2nd millennium CE), while EVI dates to the Ethio-Sabaean period with the lowest layer dating to the late 2nd millennium BC [cf. 26]. These characteristics lead to notable differences between the two profiles. There are perceptible variations of $PO_4$, $Fe_2O_3$, soil organic matter (SOM) and $CaCO_3$ contents throughout Profile EI (S1 Fig); otherwise EI is a low contrast profile with gradual change occurring in most of the examined parameters. The calculated weathering indices applied indicate minor differences between the layers examined in the present study but capture the overall structure of the stratigraphy (Table 1). Grain size composition only shows minor variation throughout profile EI. Contrary to this, Profile EVI shows more distinct differences between its layers (Fig 2, Table 1). Due to the strong variation of $PO_4$, $Fe_2O_3$, SOM and $CaCO_3$ contents as well as weathering indices and grain size composition, Profile EVI must be regarded as a high contrast profile with multiple layers that differ significantly in colour and overall brightness (i.e. spectral properties), the most prominent layers are EVI-B1, EVI-B2 and EVI-D. Detailed results of the sedimentological analyses carried out for profiles EI and EVI are shown in Table 1 and S1 Fig. Both profiles additionally differ in their situation within the trench: profile EI is part of a free-standing profile wall that is easily accessible whereas Profile EVI is situated in a narrow part of the trench where orthogonal profile walls are present directly left and right of the areas examined (Fig 2).

### Materials and methods

Our methodological approach includes (i) the laboratory analysis of chemical and mineralogical components as well as grain size composition of the sediment samples, (ii) the acquisition of multi- and hyperspectral image data, (iii) random forest regression analyses and predictions

**Table 1. Stratigraphic layers of profiles EI and EVI.** Selected sediment properties obtained from the sampled materials (KT: K-Ti-ratio; RX: Ruxton-ratio). Depth of samples given in cm below the modern surface. See S1 Fig and S1 and S2 Files for the detailed sedimentological record.

| Layer | sample no. | sample depth [cm] | Description | SOM [mass%] | Fe₂O₃ [mass%] | CaCO₃ [mass%] | KT | RX |
|---|---|---|---|---|---|---|---|---|
| **Profile EI** | | | | | | | | |
| Layer | sample no. | sample depth [cm] | Description | SOM [mass%] | $Fe_2O_3$ [mass%] | $CaCO_3$ [mass%] | KT | RX |
| EI-A | 7 | 184 | anthropogenic layer; grey-brown silty loam; inclusions of ash, charcoal, bones and ceramics | 4.7 | 9.3 | 3.9 | 37.1 | 6 |
| EI-B | 6 | 200 | anthropogenic layer; grey-brown silty loam; few inclusions; homogeneous | 5.1 | 10.2 | 4.3 | 30.4 | 5.4 |
| EI-C | 5 | 210 | anthropogenic layer; compacted brown-grey silty loam; inclusions of charcoal, stones, bones and ceramics | 2.2 | 8.9 | 6.2 | 31.7 | 5.9 |
| EI-D | 4 | 221 | anthropogenic layer; dark grey and brown silty loam with a high content of charcoal and ash; some bones and ceramics | 3.9 | 10.2 | 3.6 | 27.9 | 6.2 |
| EI-D1 | 3 | 228 | a concentration of darker and more brown silty loam inside layer D | 3.2 | 11.3 | 2.6 | 27.9 | 6 |
| EI-E | 2 | 240 | compacted silty loam yellow-brown sediment with very few inclusions; high loam content | 2.6 | 12.9 | 1.6 | 27.6 | 5.7 |
| EI-F | 1 | 257 | anthropogenic layer; compacted brown-grey silty loam with local clusters of small stones; high loam content; some bones and ceramics, some charcoal | 2.1 | 13.4 | 2.1 | 26.3 | 5.2 |
| **Profile EVI** | | | | | | | | |
| Layer | sample no. | sample depth [cm] | Description | SOM [mass%] | $Fe_2O_3$ [mass%] | $CaCO_3$ [mass%] | KT | RX |
| EVI-A | - | - | a thick layer of broken stones (supposedly deposited intentionally); separates the examined section from overlying stratigraphy | - | - | - | - | - |
| EVI-B | 13 | 682 | anthropogenic layer; dark brown-grey silty loam; inclusions of charcoal, ash, bones, ceramics, and burnt clay | 1.4 | 14.1 | 1.5 | 13.6 | 4.2 |
| EVI-B1 | 15 | 657 | grey to white bands of ash scattered throughout layer B; partly with charcoal | 3.5 | 8 | 28 | 12.9 | 5.6 |
| EVI-B2 | 14 | 670 | black charcoal layer; partly with burnt clay | 7.5 | 12.3 | 10.3 | 13.7 | 4.7 |
| EVI-C | 10 | 729 | anthropogenic layer; compacted brown-grey sandy loam; inclusions of burnt clay, bones and ceramics | 0.9 | 13.9 | 2.2 | 17.9 | 4.7 |
| EVI-C1 | 11 | 734 | red-grey homogeneous feature of silty loam | 1.3 | 14.7 | 2.9 | 20.2 | 5.1 |
| EVI-D | 9 | 742 | homogeneous feature of grey ash | 1.2 | 8.4 | 14.5 | 30.9 | 4.9 |
| EVI-E | 8 | 757 | anthropogenic layer; dark brown-grey loam; inclusions of bones and ceramics | 0.2 | 19.4 | 5.5 | 15.8 | 5.1 |

of the examined sediment parameters with the spectral data received from the images, and (iv) accuracy assessment of the results. Our experimental design draws on common practice during excavations: we examined selected areas of two archaeological sections of similar size (c. 1.4 x 1.0 m, Figs 1 and 2) where we extracted one or two sediment samples from each stratigraphic layer, resulting in a total number of 16 samples for both profiles together. We thank the Authority for Research and Conservation of Cultural Heritage (ARCCH) and the Tigray Culture and Tourism Bureau (TCTB) for permission to take samples and analyse them in Germany. Due to the observed differences between the two profiles analysed, regression analyses were individually performed for each profile. To ensure transparency of our results, our data is available in the (S1 and S2 Files) along with scripts for the programming language R.

## Laboratory analysis

We analysed 16 sediment samples, extracted from the two profiles EI and EVI. Additionally one sample of the underlying saprolite was analysed, extracted from the weathered parent material lying beneath profile EVI. Sampling locations are shown in Fig 2. The samples were analysed in the laboratory of Physical Geography, Freie Universität Berlin.

The water content of the sediments was determined gravimetrically, calculated according to Blume et al. 2011 [28] and reported in mass%. The particle size distribution (1mm–0.04$\mu$m) of

the sampled material was determined with a laser diffraction particle size analyser (Beckmann-Coulter LS13 320): the prepared samples were put into a liquid sample divider and two sub-samples were measured with three independent runs each. The six measurements per sample were averaged to obtain the sample's grain size distribution [29]. Particle sizes are defined according to Ad-hoc Arbeitsgruppe Boden 2005 [30] and reported in vol%. Analysis of element contents was conducted with (i) a Thermo Scientific Niton XL3t portable energy-dispersive X-ray fluorescence spectrometer (p-ED-XRF) and (ii) a PerkinElmer Optima 2100 DV inductively coupled plasma optical emission spectrometer (ICP-OES) after aqua regia digestion. For later statistical analyses element concentrations of Si, K, Mg and Ca measured by p-ED-XRF were used, while Na, Fe, Al, Ti and $PO_4$ concentrations were used as provided by ICP-OES data (S1 File). Several certified reference materials (CRM) were applied for quality control: NCS DC 73325 (soil), NCS DC 73387 (soil), NCS DC 73389 (soil), LKSD-2 (lake sediment), LGC6156 (harbour sediment) and LGC6180 (flue ash). Carbon contents of the samples were examined for total carbon content (TC mass%) by burning at 1000˚C in an oxygen flow (LECO TruspecCHN + S-Add-On) and inorganic carbon content (TIC mass%) by the evolution of $CO_2$ during acid ($H_3PO_4$) treatment and the subsequent quantification of the evolved $CO_2$ in 20 ml 0.05 N NaOH solution by conductivity (Woesthoff Carmhograph C-16); total organic carbon content (TOC, reported as SOM in mass%) was calculated by subtracting TIC from TC. Calcite ($CaCO_3$) contents were calculated from the measured TIC values. Mineral composition of the samples was examined with a Rigaku MiniFlex 600 X-ray powder diffractometer (XRD) with a copper $k\alpha$ tube. Mineral presence was examined semi-quantitatively using the software Philips X'Pert HighScore (v. 1.0b).

An ASD FieldSpec II spectroradiometer was used in the laboratory to capture visible (VIS) and near-infrared (NIR) reflectance of the sediment samples extracted from the profiles (350 nm–2500 nm, 1 nm steps). Samples were measured before and after homogenisation. Each sample was illuminated by two halogen lamps. Variation of the results was minimised by capturing a white reference every 15 minutes. Each spectrum was averaged from 50 single measurements to compensate for uneven sediment texture. The captured spectra were smoothed using a Savitzky-Golay filter of second polynomial order with a width of 21 values [31, 32]. The processed spectral data was used as a reference for the hyperspectral image data acquired during fieldwork.

## Weathering indices

We used the results of the chemical sediment analyses to calculate multiple weathering indices (Table 2) applying ratios of certain elements or minerals to highlight differences between stratigraphic layers. Recently Zhang and Hartemink [12, 33] showed that certain weathering indices have proven useful for the delineation of high contrast soil horizons based on image data: they observed good performance of Ca-Ti-ratio [CTR; 34], the Ruxton-ratio [RX; 35] and the Sesquioxide-ratio [SQ; 35]. We also calculated these indices to test if their results can be

**Table 2. Weathering indices used throughout the study.** The weathering index after Parker (1970) was calculated using the atomic proportion, defined as the atomic percentage divided by atomic weight.

| Weathering Indices | Formula | Literature |
|---|---|---|
| Ruxton Ratio (RX) | $Si_2O_3 / Al_2O_3$ | Ruxton 1968 |
| Sesquioxide Ratio (SQ) | $Si_2O_3 / (Al_2O_3 + Fe_2O_3)$ | Ruxton 1968 |
| Calcium/Titanium Ratio (CTR) | $CaO / TiO_2$ | e.g. Betard 2012 |
| Potassium/Titanium Ratio (KT) | $K / Ti$ | e.g. Davies et al. 2015 |
| Parker's Index (PI) | $[(Na^*/0.35) + (Mg^*/0.9) + (K^*/0.25) + (Ca^*/0.7)] \times 100$ | Parker 1970 |

**Table 3. Technical specifications of the camera systems used throughout this study.**

|  | spatial resolution | spectral resolution | colour depth |
|---|---|---|---|
| Sony A6000 | 5696 x 4272 px | RGB in 3 bands | 8 bit |
| Cubert X2 S258 | 512 x 272 px | 450(480)—850(830) nm in 41 (36) bands | 16 bit |

transferred to complex archaeological profiles. Additionally we examined the K-Ti-ratio (KT) due to the high difference in ionic potential of both factors [cf. 36, 37]. Furthermore we included Parker's index [PI; 38] as a widely accepted standard [e.g. 39, 40]. Calculations of all indices include the molar mass of the respective elements.

## Image acquisition

Image acquisition differs slightly from the setup described by Haburaj et al. 2019 [5]. Digital RGB photographs of the sections were taken using a 24.3 MP mirrorless camera with a 2.8/30 mm lens (Sony ILCE-6000, Sigma 30mm F2.8 DN Art). Hyperspectral imaging was conducted using a Cubert X2 S258 snapshot camera with a Schneider-Kreuznach Cinegon 1.8/16 mm lens. Technical specifications of the camera systems are given in Table 3.

The images were captured using halogen lighting (500W) to obtain uniform lighting. In total 13 hyperspectral images were recorded for profile EI and 12 hyperspectral images for profile EVI. Every single image was processed with a separate white reference measurement, allowing us to increase quality of the spectral data by including spatial variations in the lighting conditions. Additionally, each profile was captured by a single RGB image taken with the Sony ILCE-6000 camera. Black and white reference targets were used for orthorectification of RGB and hyperspectral images and the creation of overlapping data for each profile.

Hyperspectral images were recorded in 16-bit TIF-format using the proprietary software Cubert Utils Touch. The 8-bit images of the Sony ILCE-6000 RGB camera were also converted to TIF-format. For further processing, the pixel values of both cameras were normalised to the range [0, 1] via feature-scaling and the respective colour depth.

Since the hyperspectral images showed strong vignetting, the corners of all recordings were removed by cropping, resulting in a spatial resolution of 428 x 254 pixels. The images were stitched manually in QGIS (v3.4), using thin plate spline (TPS) transformation and nearest neighbour (NN) resampling and including the rectified RGB images as spatial reference. Histogram matching (R package RStoolbox, [41]) was used to eliminate the remaining differences between single images. Images were then merged using GDAL (v2.4.1).

The VIS-NIR spectral data recorded with the ASD spectroradiometer for each sediment sample was used as a reference for the correction of the hyperspectral image data. A direct comparison of the two spectral datasets revealed that additional steps were necessary to eliminate noise in the image data (Fig 3). The merged hyperspectral image data was spatially filtered using a 5x5 median filter. The image bands from 560 to 670 nm (n = 12) show high noise content since in this spectral range the measured spectrum is a composite from two overlapping sensors of the camera system (VIS sensor and NIR sensor). This range was masked and interpolated using piecewise cubic hermite interpolating polynomial interpolation (R package signal, [42]). The spectra were then smoothed with a Savitzky-Golay filter of second polynomial order with a width of 11 values (Savitzky and Golay 1964). The image bands representing 450, 460, 470, 840 and 850 nm were excluded from the data due to their high noise content, leading to a final spectral range of 480–830 nm in 10 nm steps. Image cells containing missing values were interpolated using a 9x9 median filter.

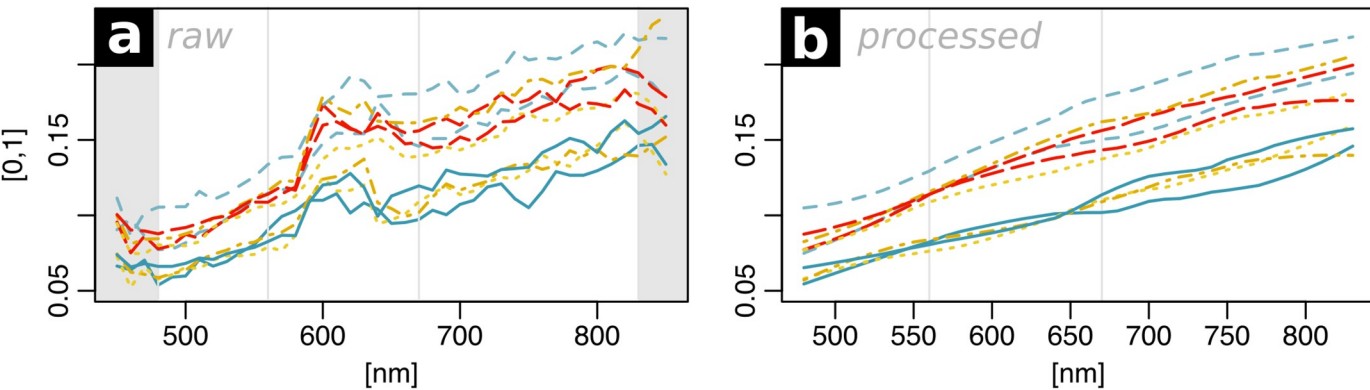

**Fig 3. Spectral data before (a) and after (b) denoising.** The signal was limited to the wavelength range of 480–830 nm and the bands between 560 and 670 nm were interpolated. The plot shows ten randomly selected spectra.

A PTFE coated white reference was used for all hyperspectral recordings. The spectrum of the white reference was additionally captured using an ASD FieldSpec Handheld 2 spectrora-diometer (325–1075 nm, 1 nm steps). The spectrum was adjusted to the spectral resolution of the hyperspectral camera and all camera recordings were divided by that spectrum to ensure accurate measurements.

To create a uniform spatial resolution as a basis for image analyses, the RGB data was downsampled to match the resolution of the hyperspectral data. New pixel values were generated by applying a mean filter. The black and white reference targets were masked manually and remaining strong shadows were masked using threshold values for the image bands CIE $b^*$ and 830 nm. This allowed us to exclude this data from the subsequent image analyses.

## Mapping of sediment properties

Random forest regression analysis was used for the spatial mapping of sediment properties using the results of the laboratory analyses and the image data.

Various studies suggest the transformation of spectral data prior to processing [13, 43–46]. We therefore created the following datasets as input for the regression analyses: (i) the RGB data, (ii) the CIELAB data derived from the RGB data, and (iii) the pre-processed hyperspectral data (480–830 nm, 10 nm steps).

Training areas for the prediction of sediment characteristics were indicated by polygons marking the sampling areas. Regression models (x,y) were trained using the spectral information from pixels (= pixel values covered by polygons) as predictors (x) and the respective sedimentological data (obtained from the laboratory analyses) as response values (y). In total we sampled c. 0.56% of the pixels of profile EI and c. 0.28% of the pixels of profile EVI. This led to a total of 3,581 hyperspectral and 3,455 RGB and CIELAB pixels for the seven samples of profile EI; correspondingly 2,054 hyperspectral and 1,395 RGB and CIELAB pixels for the nine samples of profile EVI were available.

Regression models were trained for profiles EI and EVI separately based on the textural, chemical and spectral characteristics of the respective sediment samples using a random forest algorithm after Breiman 2001 [47], (R package randomForest, [48]). Regression analyses were applied to SOM, $Fe_2O_3$, $CaCO_3$, the weathering indices SQ, RX, PI, KT and CTR, as well as the classified grain size (clay, silt and sand) as dependent variables and the spectral data as independent variables.

We used all sampled pixels as training data and performed a leave-p-out cross-validation with p = 0.3 over 50 iterations (R package rfUtilities [49]). This allowed us to verify the out-of-bag errors of the random forest regressions and to report mean and standard deviation of the root mean square error (RMSE), mean bias error (MBE) and pseudo $R^2$ values (1 − $MSE/Var(y)$) from the cross-validation of each model [48]. These values were used for quality assessment of the results along with the on-site delineation of the profiles conducted by archaeological experts. The regression models were used for prediction with the image data and the R package raster [50].

## Results

The deposits exposed in profiles EI and EIV correspond to a silty to sandy loose sediment, brownish to greyish in colour and with quartz and feldspar as the predominating mineral components. Individual stratigraphic features are characterised by compaction material, ash inclusions or varying amounts of clay (Table 1, Fig 2, S1 Fig). In addition, XRD data emphasises that most samples are characterised by distinct hematite ($Fe_2O_3$) contents (S1 File); as Fe-bearing minerals other than hematite are circumstantial along the two sediment profiles, for the ongoing analysis the measured Fe contents are recognised as hematite components. The underlying saprolite has developed from the parent granitic bedrock and shows low carbon contents (TC: 1.55 mass%; TIC: 1.11 mass%; TOC: 0.44 mass%). Low carbonate contents all along the profiles correspond to relatively low calcium concentration, which is due to the predominance of granites in the drainage basin. Peaks of carbonate (> 10 mass%) only occur in the stratigraphic features EVI-B1, EVI-B2 and EVI-C1 in profile EVI. $PO_4$ concentrations along both profiles are low—slightly increased only in layers EI-A, EI-C, EI-D and EI-D1 in profile EI and showing a singular peak in layer EVI-D in profile EVI (c. 3-4.5 vol%).

The initially described differences between the two profiles are also visible in their sedimentological record: minor changes of the examined parameters are visible throughout profile EI (Table 1), where layer EI-C shows the most distinct differences to its overlying and underlying layers in $PO_4$, SOM and $CaCO_3$ contents. Most other changes along profile EI are gradual. In contrast, in profile EVI we observed more pronounced differences between the individual layers in water content, SOM, $PO_4$, $Fe_2O_3$, $CaCO_3$ contents and the calculated weathering indices.

We trained random forest regression models between the spectral data acquired for the sampling points from the acquired image data and the examined sediment parameters (SOM, $Fe_2O_3$, $CaCO_3$, the weathering indices SQ, RX, PI, KT, CTR, classified grain size). Thereby, we were able to predict maps of these parameters which cover profiles EI and EVI. Evaluation of the trained models was carried out by leave-p-out cross-validation (Table 4). Only prediction maps with an $R^2$ value greater than 0.8, low values of RMSE and standard deviation and a good correlation of the prediction results with the on-site stratigraphic interpretation are evaluated as significant and therefore depicted in Figs 4–7. To ensure transparency, all produced models and maps are available in the S1 File.

### Profile EI

We produced maps of multiple sediment parameters throughout profile EI by applying statistical models based on RGB and hyperspectral data. The RGB data of profile EI produced good results when predicting SOM (Figs 4 and 6a, Table 4; $R^2$: 0.9). Similar results were obtained applying the RGB derived CIELAB data (Fig 6b; $R^2$: 0.93). Both sets of image data managed to roughly capture the main differences between the most prominent layers of profile EI: the increase in SOM content of layers EI-A, EI-B and EI-D. The RGB derived CIELAB data additionally captured the differences in silt and sand concentration throughout profile EI,

**Table 4. Cross-validation results of the regression analyses using RGB, CIELAB and hyperspectral data.** Mean and standard deviation of the root mean square error (RMSE), mean bias error (MBE) and pseudo $R^2$ values ($1 - MSE/Var(y)$) obtained from leave-p-out cross-validation are reported. The training data consisted of seven sediment samples for profile EI and nine sediment samples for profile EVI. Significant models are marked in bold.

| | | Profile EI (n = 7) | | | | | | Profile EVI (n = 9) | | | | | |
|---|---|---|---|---|---|---|---|---|---|---|---|---|---|
| | | $R^2$ | $R^2(sd)$ | RMSE | RMSE (sd) | MBE | MBE (sd) | $R^2$ | $R^2(sd)$ | RMSE | RMSE (sd) | MBE | MBE (sd) |
| SOM | RGB | **0.9** | 0.003 | 0.366 | 0.012 | 0.001 | 0.012 | **0.961** | 0.004 | 0.384 | 0.079 | -0.018 | 0.021 |
| | CIELAB | **0.932** | 0.002 | 0.301 | 0.01 | -0.001 | 0.011 | **0.987** | 0.002 | 0.214 | 0.065 | -0.005 | 0.014 |
| | 480–830 nm | **0.883** | 0.005 | 0.385 | 0.021 | 0.004 | 0.015 | **0.933** | 0.006 | 0.503 | 0.069 | 0.006 | 0.023 |
| $Fe_2O_3$ | RGB | 0.446 | 0.015 | 1.188 | 0.03 | 0.004 | 0.044 | **0.942** | 0.006 | 0.837 | 0.089 | -0.002 | 0.057 |
| | CIELAB | 0.599 | 0.013 | 1.016 | 0.032 | 0.002 | 0.039 | **0.974** | 0.003 | 0.556 | 0.092 | -0.001 | 0.034 |
| | 480–830 nm | **0.886** | 0.003 | 0.54 | 0.022 | -0.007 | 0.02 | 0.925 | 0.003 | 0.972 | 0.067 | -0.016 | 0.05 |
| $CaCO_3$ | RGB | 0.324 | 0.016 | 1.265 | 0.03 | -0.008 | 0.047 | **0.972** | 0.003 | 1.016 | 0.185 | -0.032 | 0.057 |
| | CIELAB | 0.507 | 0.015 | 1.079 | 0.032 | -0.003 | 0.038 | **0.989** | 0.002 | 0.678 | 0.195 | -0.026 | 0.043 |
| | 480–830 nm | **0.822** | 0.006 | 0.649 | 0.027 | 0.004 | 0.026 | 0.965 | 0.003 | 1.113 | 0.132 | -0.009 | 0.047 |
| RX | RGB | 0.53 | 0.011 | 0.21 | 0.004 | 0 | 0.008 | **0.945** | 0.004 | 0.106 | 0.011 | -0.001 | 0.007 |
| | CIELAB | 0.663 | 0.009 | 0.176 | 0.005 | 0 | 0.007 | **0.978** | 0.002 | 0.065 | 0.011 | 0.001 | 0.004 |
| | 480–830 nm | **0.828** | 0.004 | 0.124 | 0.004 | 0.001 | 0.005 | 0.875 | 0.01 | 0.162 | 0.017 | -0.004 | 0.008 |
| SQ | RGB | 0.457 | 0.014 | 0.18 | 0.005 | -0.001 | 0.008 | **0.973** | 0.003 | 0.058 | 0.006 | -0.002 | 0.004 |
| | CIELAB | 0.602 | 0.012 | 0.151 | 0.005 | 0 | 0.006 | **0.988** | 0.001 | 0.038 | 0.006 | 0 | 0.002 |
| | 480–830 nm | **0.897** | 0.003 | 0.076 | 0.003 | 0.001 | 0.003 | 0.97 | 0.001 | 0.059 | 0.004 | 0 | 0.003 |
| CTR | RGB | 0.318 | 0.015 | 12.619 | 0.298 | -0.126 | 0.499 | **0.98** | 0.004 | 5.127 | 1.69 | -0.128 | 0.297 |
| | CIELAB | 0.502 | 0.011 | 10.716 | 0.242 | -0.015 | 0.45 | **0.99** | 0.003 | 4.437 | 1.189 | -0.033 | 0.319 |
| | 480–830 nm | **0.847** | 0.005 | 5.938 | 0.25 | 0.066 | 0.239 | 0.99 | 0.001 | 3.777 | 0.778 | -0.058 | 0.177 |
| KT | RGB | 0.605 | 0.011 | 2.17 | 0.047 | -0.027 | 0.066 | **0.961** | 0.004 | 1.147 | 0.202 | 0.045 | 0.071 |
| | CIELAB | 0.726 | 0.008 | 1.809 | 0.045 | 0.006 | 0.066 | **0.985** | 0.003 | 0.68 | 0.217 | -0.015 | 0.045 |
| | 480–830 nm | **0.861** | 0.004 | 1.266 | 0.052 | 0.021 | 0.047 | 0.951 | 0.004 | 1.246 | 0.209 | -0.008 | 0.046 |
| PI | RGB | 0.478 | 0.013 | 6.9 | 0.141 | -0.035 | 0.271 | **0.987** | 0.002 | 2.782 | 0.364 | -0.102 | 0.196 |
| | CIELAB | 0.62 | 0.013 | 5.878 | 0.164 | 0.016 | 0.217 | **0.994** | 0.001 | 2.089 | 0.51 | -0.092 | 0.126 |
| | 480–830 nm | **0.876** | 0.003 | 3.335 | 0.156 | 0.007 | 0.115 | **0.99** | 0.001 | 2.574 | 0.435 | -0.044 | 0.145 |
| clay | RGB | 0.542 | 0.014 | 1.033 | 0.034 | 0.004 | 0.04 | 0.837 | 0.017 | 1.911 | 0.2 | -0.041 | 0.122 |
| | CIELAB | 0.661 | 0.012 | 0.897 | 0.035 | -0.006 | 0.034 | 0.929 | 0.008 | 1.271 | 0.265 | -0.041 | 0.071 |
| | 480–830 nm | **0.901** | 0.003 | 0.48 | 0.023 | -0.003 | 0.018 | 0.959 | 0.003 | 0.955 | 0.089 | -0.012 | 0.043 |
| silt | RGB | 0.787 | 0.008 | 1.893 | 0.058 | -0.014 | 0.057 | 0.934 | 0.004 | 3.216 | 0.337 | -0.096 | 0.182 |
| | CIELAB | **0.862** | 0.006 | 1.502 | 0.063 | 0.011 | 0.055 | 0.976 | 0.001 | 2.018 | 0.294 | -0.05 | 0.128 |
| | 480–830 nm | **0.834** | 0.006 | 1.665 | 0.09 | 0.014 | 0.061 | 0.898 | 0.005 | 4.009 | 0.337 | -0.038 | 0.175 |
| sand | RGB | 0.765 | 0.008 | 2.21 | 0.06 | 0.02 | 0.082 | 0.904 | 0.007 | 4.597 | 0.395 | 0.079 | 0.262 |
| | CIELAB | **0.838** | 0.006 | 1.834 | 0.072 | -0.005 | 0.076 | 0.965 | 0.003 | 2.742 | 0.444 | 0.084 | 0.21 |
| | 480–830 nm | **0.81** | 0.006 | 1.961 | 0.095 | 0.018 | 0.072 | 0.901 | 0.004 | 4.682 | 0.344 | 0.07 | 0.278 |

especially visible in layer EI-E (Fig 6l and 6n). Cross-validation results for these parameters are robust with high $R^2$ and low RMSE values as well as low standard deviations (Table 4). These values are significantly lower for the remaining sediment parameters of profile EI. Prediction results of weathering indices, $Fe_2O_3$, $CaCO_3$ and clay, as calculated from RGB data and RGB derived CIELAB data of profile EI are therefore not reliable (Table 4).

The hyperspectral image data of profile EI generally produced more accurate results for all of the predicted parameters when compared to the results obtained by applying the RGB data and the RGB derived CIELAB data. In general, we observed a good congruence between calculated and on-site delineated maps by visual comparison. Additionally, regression models calculated using the hyperspectral data of profile EI and textural and chemical sediment

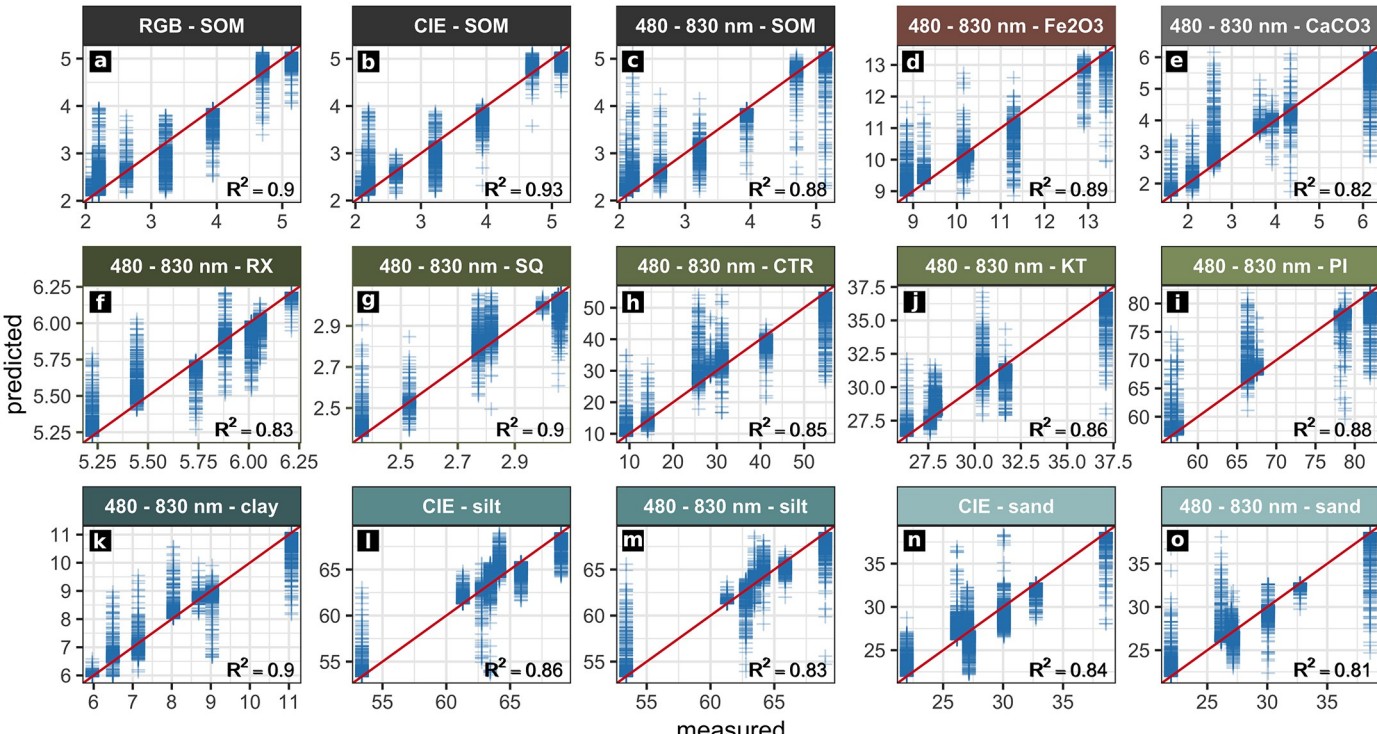

**Fig 4. Results from regression analyses of profile EI.** Actual (measured) and predicted values of the sediment properties examined. The ideal performance is shown by the red line (y = x). Only models with a cross-validated $R^2$ greater than 0.8 and a high correlation between the respective prediction map and the stratigraphic delineation are shown.

characters as independent variables produced $R^2$ values between 0.81 and 0.9 (n = 7) and low error values (Table 4). These statistical results document the reliability of the parameter maps produced by these models and underline their good agreement with the on-site delineation. The most reliable models generated from the hyperspectral data of profile EI were derived for SOM contents, hematite ($Fe_2O_3$) contents, the SQ weathering index and clay contents (Fig 6c, 6d, 6g and 6k). Applying these statistical models based on the hyperspectal image data, layers EI-E and EI-F were successfully separated from the rest of the profile regarding their $Fe_2O_3$ and $CaCO_3$ contents, grain size composition and weathering indices (Fig 6). Additionally, layers EI-D and EI-D1 were reliably outlined by applying models of the weathering index RX, SOM content and grain size composition based on hyperspectal image data. The predicted map of SOM successfully captures the high values present throughout layers EI-A and EI-B (Fig 6c, S1 Fig). These two layers also show a low sand content in the sedimentological record (S1 Fig) and the parameter maps (Fig 6o).

## Profile EVI

The random forest regression models and the resulting prediction maps produced using the RGB image data of profile EVI show good results for SOM, $Fe_2O_3$ and $CaCO_3$ contents, as well as for the weathering indices SQ, CTR, KT and PI (Figs 5 and 7, Table 4). Increased $Fe_2O_3$ contents for layers EVI-C1 and EVI-E are clearly visible in the map based on the RGB data (Fig 7d). The black charcoal layer in EVI-B2 is clearly captured by the prediction map of SOM derived from the RGB data (Fig 7a). Prediction maps for profile EVI based on regression models calculated for the RGB derivative CIELAB are similar to those produced by RGB data

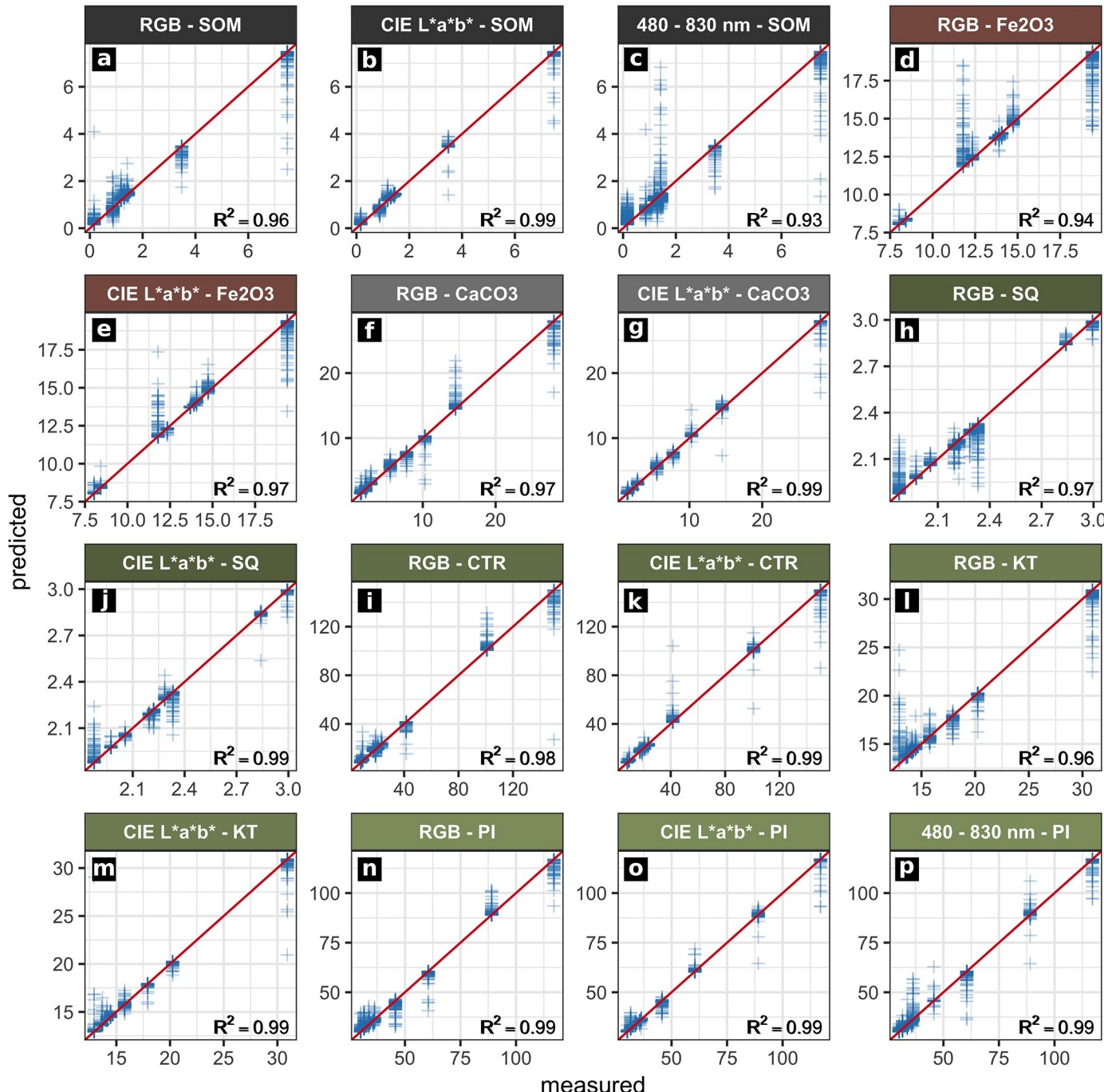

**Fig 5. Results from regression analyses of profile EVI.** Actual (measured) and predicted values of the sediment properties examined. The ideal performance is shown by the red line (y = x). Only models with a cross-validated $R^2$ greater than 0.8 and a high correlation between the respective prediction map and the stratigraphic delineation are shown.

(Fig 7). Also the $R^2$ values for both datasets are very similar for the mentioned parameters (Table 4; $R^2 > 0.94$, n = 9). For both RGB and CIELAB data, statistical models for all weathering indices except RX outline layers EVI-B1, EVI-B2 and EVI-D (Fig 7h–7p). Mapping results based on modelling grain size composition (sand, silt, clay) are less pronounced than observed

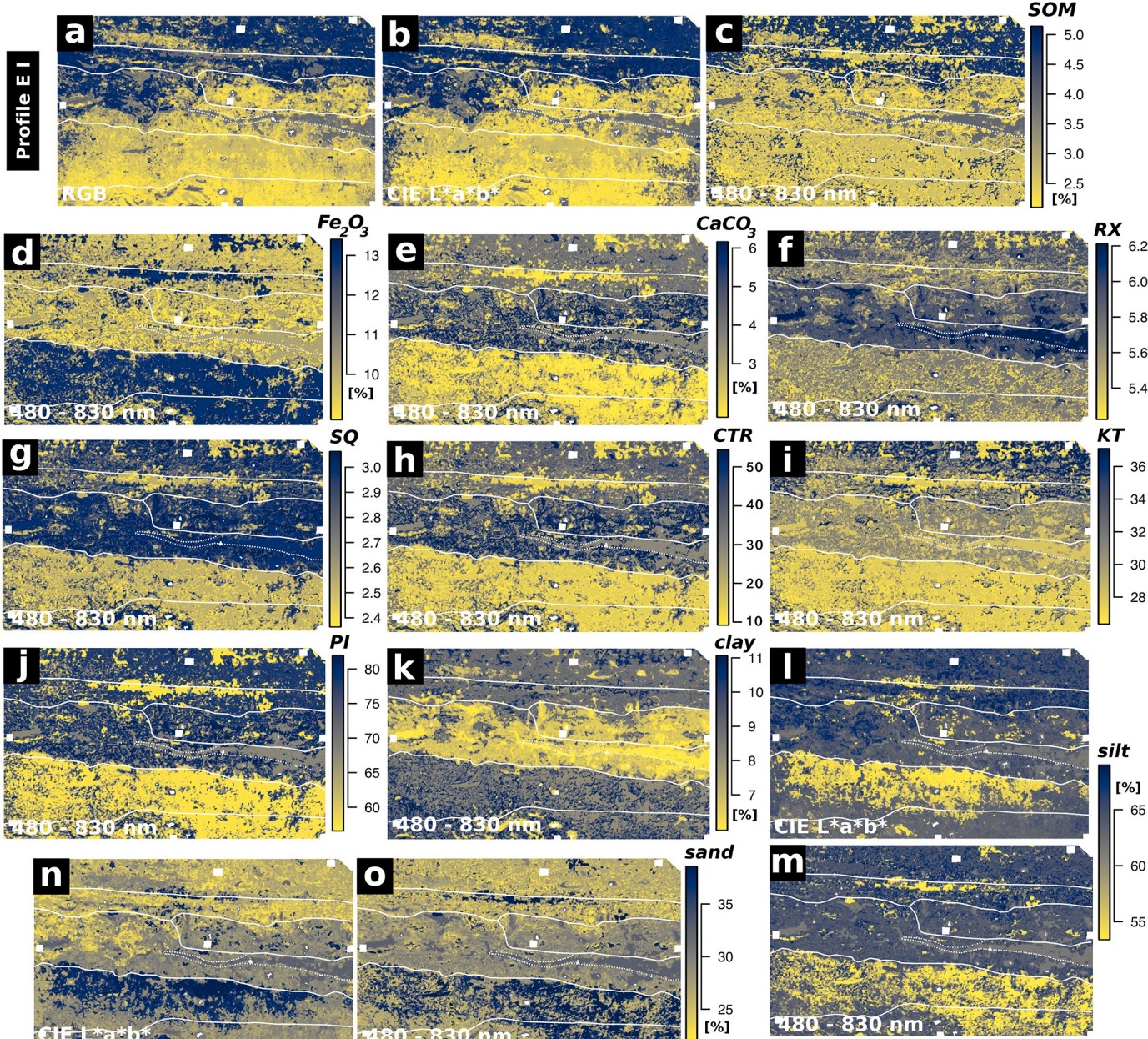

**Fig 6. Predicted sediment properties of Profile EI.** Only models with a cross-validated $R^2$ greater than 0.8 and a high correlation between the respective prediction map and the on-site stratigraphic delineation (white lines) are shown.

for profile EI; statistical models involving grain size composition and the RGB data and RGB derived CIELAB data of profile EVI are not significant (Table 4). In contrast, the $R^2$ values of modelled grain size composition based on the hyperspectral image data are greater than 0.8. However, the resulting prediction maps for profile EVI do not match with the on-site stratigraphic interpretation (S2 File).

Prediction maps generated by applying statistical models which involve the datasets from the hyperspectral images of profile EVI show a strong variance of the data inside the profile from left to right, indicating a systematic error; this shift was not captured by the on-site

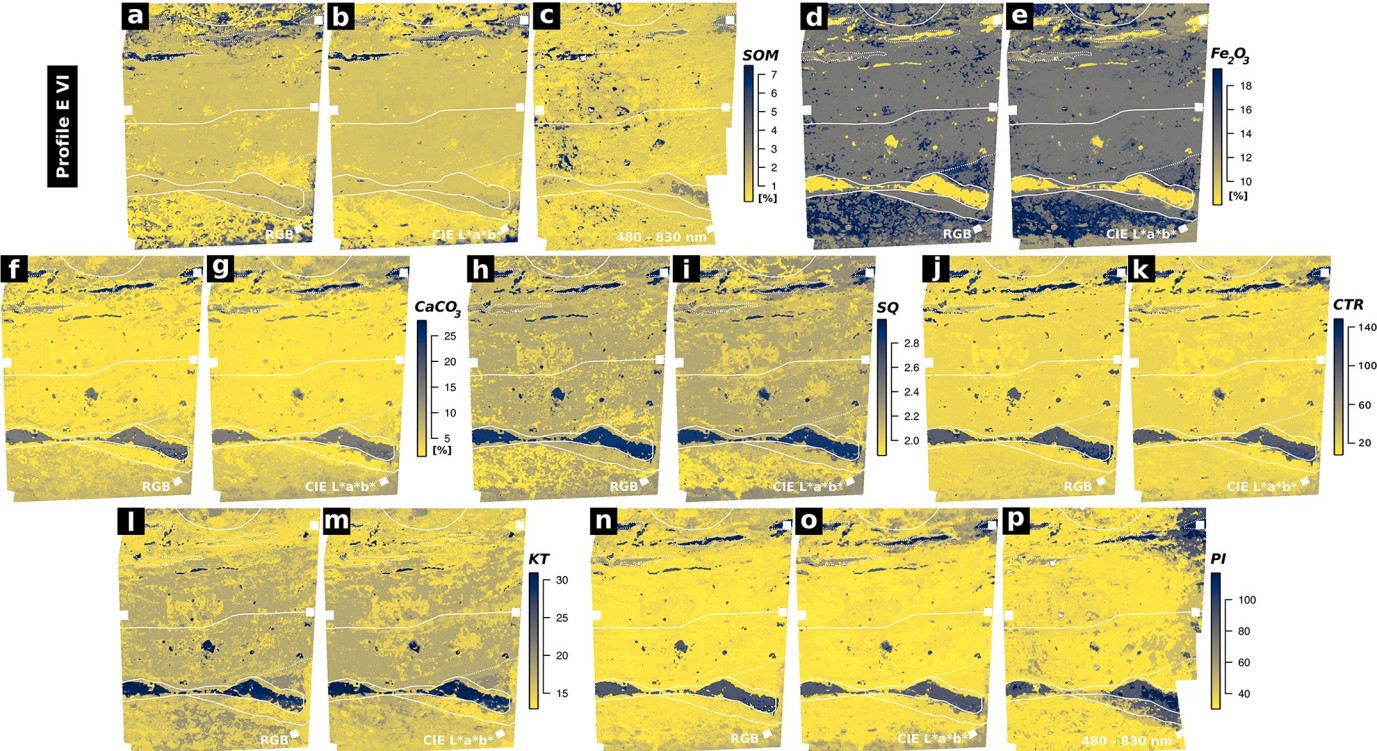

**Fig 7. Predicted sediment properties of Profile EVI.** Only models with a cross-validated $R^2$ greater than 0.8 and a high correlation between the respective prediction map and the on-site stratigraphic delineation (white lines) are shown.

stratigraphic interpretation (Figs 2 and 7). Models resulting from random forest regression analyses of the datasets from hyperspectral images and the different sedimentological parameters generally show significant $R^2$ values ($> 0.87$, n = 9, Table 4). In contrast, parameter maps produced by applying these statistical models to the hyperspectral image data only agree in part with the maps derived from the respective statistical models based on the RGB data and the RGB and its derivatives: while for the maps derived from the application of the statistical models based on RGB data and RGB derived CIELAB data the layers outlined highly correspond with the on-site delineated stratigraphy, this does not pertain to the maps generated by the application of statistical models based on hyperspectral image data. For example, this applies to the SOM contents as shown in Fig 7c: SOM concentrations are generally low and layers EVI-B1 and EVI-B2 are outlined correctly with peaking SOM contents; in addition, noise in the data does not allow a reliable delineation of layers EVI-C1 and EVI-E. Apart from this SOM map of limited quality, a map of acceptable quality based on the hyperspectral data of profile EVI could only be produced for the weathering index PI (Figs 5p and 7p). All other statistical models of the hyperspectral data of profile EVI and the remaining parameters produced good statistical significances (Table 4) but the derived parameter maps did not hold any significant information.

## Discussion

### Prediction results

**General.** Modelling layer delineation for profiles EI and EVI generally shows a good performance of models based on RGB image data for predicting soil organic matter (SOM). Layer

delineation by applying models based on the RGB derived CIELAB data leads to more homogeneous prediction maps of SOM contents (Figs 6a, 6b, 7a and 7b), which is consistent with the findings of Zhang and Hartemink 2019 [12]. The good performance of the statistical models based on hyperspectral data of profile EI is consistent with the laboratory studies of, for example, Steffens and Buddenbaum 2012, Schreiner et al 2012, Hobley et al. 2018, Heil et al. 2020 [6, 7, 51, 52] and on-site studies like Zhang and Hartemink 2019 [12]. Applying the hyperspectral data allowed us to develop statistical models which are reliably able to map the predicted layers based on the sediment properties for profile EI. In contrast, spectral noise and disturbances of the hyperspectral data of profile EVI generated relatively unreliable results for various sediment characteristics and thus did not delineate stratigraphic layers convincingly.

**SOM**. The good results for predicting SOM from all the used image data are accounted for by the fact that the visual reflectance of soils and sediments (i.e. brightness) is highly related to SOM content [cf. 53]. The observed increase in mapping quality of SOM when using the hyperspectral image data of profile EI as independent variable in the statistical models shows the potential of a higher spectral resolution throughout the visible part of the electromagnetic spectrum. Likewise the results are certainly influenced by the range of the hyperspectral sensor: the equidistant measurement of the area between 600 and 830 nm clearly adds important information compared to the RGB data measured with a sensor that shows rather limited capabilities beyond 650 nm [cf. 5]. However, the absolute difference between the mapping results of SOM from RGB and hyperspectral data as visible in Figs 6 and 7 is relatively low, thus stressing the potential of RGB imaging for the mapping of selected parameters, as suggested by e.g. Zhang and Hartemink 2019 [12].

**$Fe_2O_3$**. The iron oxide hematite shows specific spectral features between 510 and 620 nm [53, 54] and absorption bands at 450, 680 and 700 nm are also related to hematite [45, 53]. The hyperspectral sensor used directly measures these wavelengths while the RGB data just covers them in one summarising red image band; in consequence, application of the hyperspectal data leads to a good performance of the statistical model when mapping $Fe_2O_3$ in profile EI. However, as the results of profile EVI reveal, RGB and RGB based CIELAB image data can clearly be used to map hematite content throughout a profile, as long as the contents are high enough. Several authors prove a direct relation between iron oxide contents and redness of soils and sediments, e.g. through the colour coordinates a* and b* [55–57], which also explains the good performance of the random forest regression models based on RGB and RGB based CIELAB image data of profile EVI.

**Grain size**. We produced good results when mapping grain size throughout profile EI by applying statistical models based on the CIELAB and hyperspectral image data. The performance of CIELAB data was acceptable, but using hyperspectal data clearly led to more reliable maps (Fig 6). We see these results as indicating the potential of an extended spectral range when mapping grain size, as we were able to compensate for the lower number of samples in comparison to Zhang and Hartemink 2019 [12] by extending our spectral range to 480–830 nm. Data presented by Post and Noble 1993 [58] and later Stenberg et al. 2010 [54] indicates that the reliability of characterising textural elements based on statistical models increases when sensors covering NIR are included since many reflectance features related to grain size only occur beyond 1000 nm. Most studies dealing with the prediction of grain size through spectral data do indeed cover the VIS-NIR area up to 2500 nm or even use mid infrared data (MIR; 2,500–25,000 nm) [cf. 44].

**Weathering indices**. Weathering index mapping based on statistical models worked out nicely in many cases and proved helpful for assessing similarities and differences between layers (see below). Statistical models based on RGB and CIELAB image data were observed by Zhang and Hartemink 2019 [12] to perform well in predicting the indices SQ, RX and CTR;

this could be reproduced in part with our experimental design (Figs 6 and 7, Table 4). The mapping results of the Sesquioxide ratio based on the RGB and CIELAB image data were especially helpful for delimiting layers EVI-C1 and EVI-E in profile EVI (Fig 7). Derived from the statistical models based on the hyperspectral image data, the prediction map of the weathering index after Parker 1970 [38] showed a slight advantage over the weathering index CTR (derived from the same datasets) for mapping layers EI-B and EI-E / EI-F in profile EI (Fig 6). As the reliable prediction maps of PI and CTR are very similar in both examined profiles (Figs 6 and 7), we suggest the usage of CTR over PI since less laboratory analyses are necessary to calculate the CTR index (Table 2). Mapping results of the index KT (K-Ti-ratio) vary from those of other indices (Figs 6 and 7) and are likely related to grain size (as is also visible in Fig 6), as K and Ti concentration was shown to be correlated with clay, sand and silt fractions [e.g. 59].

## Consequences of and for the stratigraphic profiles examined

Several of the examined sediment parameters were helpful to delineate stratigraphic layers and thus provide supportive information for the on-site stratigraphic description. In profile EI, the boundary between layers EI-E and EI-F (as documented during fieldwork) is not visible in the generated prediction maps, which are assessed as reliable (Fig 6); however, the sedimentological record (Table 1, S1 Fig) also only points out slight differences between both layers. Likewise the increased stone contents and the anthropogenic remains in layer EI-F (Table 1) are not detectable with the experimental setup used. Stone inclusions (and layer delineation based on flakes of charcoal, bones or burnt clay) in general are a crucial drawback of the proposed method, as parameter mapping results depend on the sampled materials. Circumvention of this problem would require an increased number of samples, a higher spatial resolution (smaller pixel size) and an additional layer of documentation in the form of image texture or similar parameters [cf. 5]. However, based in the fines of layers EI-E and EI-F (Profile EI), both strata are delineated nicely from the rest of the profile in the parameter maps (Fig 6). Layers EI-A and EI-B (Profile EI) are mainly distinguished from each other by their differing density of inclusions (Table 1), again causing a relatively weak delineation between the two strata based on the application of statistical models applying RGB, its derivative CIELAB or hyperspectral data as independent variables (Fig 6). The on-site description of layer EI-B as 'more homogeneous' (Table 1) compared to layer EI-A, however, is underlined by a horizontal patch of homogeneous material visible in the mapping results of $Fe_2O_3$, $CaCO_3$ and PI, as derived from the hyperspectral image data by statistical modelling (Fig 6). The generally high contrast between layers EI-A and EI-B regarding weathering indices (Fig 6) supports the on-site description of the two layers as individual strata.

In Profile EVI, the examined sediment properties also provide supportive information for the on-site stratigraphic description. The parameter maps produced by applying the reliable statistical models show similarities between layers EVI-C1 and EVI-E (S1 Fig, Fig 7). Increased $Fe_2O_3$ contents and similar values of the SQ weathering index are visible for both layers in the respective parameter maps (Fig 7) and the increased iron content explains the red-brown colour of the two layers EVI-E and EVI-C1 (cf. [60], Table 1). The on-site description of layers EVI-D and EVI-B1 as similar material (bright ash layers) is not visible in our results. Rather, we observed clear differences between the two layers: the parameter maps of the weathering index KT produced by applying the statistical models including the RGB and CIELAB image data, indicate a significantly different weathering grade of layers EVI-D and EVI-B1 (Fig 7). This is also visible in the calculated KT and RX values in the sedimentological record of profile EVI (S1 Fig). Additionally the two layers clearly differ in $PO_4$ concentration (S1 Fig). Layer

EVI-B1 can clearly be described as a bright ash layer connected with anthropogenic activities. In contrast, we interpret layer EVI-D as a thin layer covering layers EVI-E and EVI-C1 and most likely related to a reshaping of the surface or construction work. Remains of this process are also visible on the opposite side of the trench at the same depth [26]. The lowermost anthropogenic layer (layer EVI-E) varies greatly in thickness and locally shows a gradual transition into the underlying saprolite [cf. 26, 61].

## Reproducibility

Evaluating our results, clear discrepancies are visible between profiles EI and EVI regarding the performance of the RGB, RGB derived CIELAB and hyperspectral data. For profile EI, using statistical models which are based on the hyperspectral data led to a general increase of the quality of the prediction results when compared to the results achieved by the RGB and the derived CIELAB data (Figs 4 and 6, Table 4). In contrast, using the hyperspectral data of profile EVI in most cases led to more inaccurate parameter maps than the maps produced on the basis of the RGB and the derived CIELAB data. Together with the irregular cross-validation results of the hyperspectral data of profile EVI (Table 4), we interpret this as resulting from the observed variations, noise and disturbances in the hyperspectral image data of profile EVI, presumably caused by light that was irregularly reflected by the adjacent profile walls between the recordings of the 12 hyperspectral images of profile EVI.

The two profiles additionally differ significantly in their overall characters: profile EI is a low contrast profile and EVI is a high contrast profile. Pronounced differences in lightness and redness are only visible in profile EVI (Fig 2). These characteristics result from a high variation of SOM contents (lightness) and $Fe_2O_3$ contents (redness) throughout the profile. As lightness and redness heavily influence the spectral reflectance in the visual part of the electromagnetic spectrum, the statistical models based on the RGB data show a better performance for profile EVI than for the low contrast profile EI (Figs 6 and 7).

Prediction results of both profiles are influenced by our sampling strategy, which we limited to less than ten sediment samples per profile. The digitised training areas for the regression analyses were focused on the direct surroundings of the samples and thus only covered a small percentage of the total image pixels ($< 0.6\%$). This low number of sediment samples leads to a shorter processing time, and thus lower costs, compared to other studies [e.g. 7, 12]. Generally, the gradual and rather low variation of the parameters examined throughout profile EI was captured successfully despite the low amount of samples (n = 7). Profile EVI, however, shows a more complex stratigraphy that probably would have required a more detailed sampling strategy to be able to outline its stratigraphic character: (i) singular lenses of very dark and very bright material, (ii) gradual variation in vertical and horizontal directions, and (iii) many anthropogenic layers which show differing contents of bones, charcoal, ceramics and burnt clay. These characteristics may lead to a disproportional representation of some of the examined sediment properties in the samples and thus in the parameter maps derived from the statistical models based on these samples. Nonetheless, we see the main reason for the poor performance of the hyperspectral data of EVI in the lighting situation during image acquisition: the adjacent walls of the trench are situated directly left and right of the captured part of profile EVI. Similar to the shadow cast by adjacent trench walls as described by Haburaj et al. 2019 [13], the walls in profile EVI most likely reflected and scattered the artificial light differently between single measurements due to changing lamp positions and caused the observed distortions in the hyperspectral image data of profile EVI.

Our results clearly show that a low number of samples can be sufficient to successfully map parameters like the concentrations of organic matter or iron oxide. Nevertheless, the samples

need to be representative for the profile examined. Complex anthropogenic layers featuring high amounts of flakes of brick, charcoal, bones or mortar can be reliably delineated with a very high spatial resolution of the image data and the sampling strategy. Apart from the contrast of the examined profile or planum, the spatial scale and structure should be kept in mind. Similar problems and thoughts have already been discussed in the field of remote sensing: additionally to the sizes and spatial relationships of the objects in the recorded scene (as suggested by [62]),the spectral relationships of these objects should be considered when assessing the required scale of a scene.

As the CIELAB image data used throughout the presented study is derived from the acquired RGB image data, it should be kept in mind that the parameter maps of the two datasets are highly correlated (e.g. Pearson correlation coefficient of 0.92 with $p < 0.05$ between SOM values predicted from RGB and CIELAB data, see S2 File). Transformation of the RGB data to the CIELAB colour space, however, has some benefits: the parameter maps based on the CIELAB data frequently showed less noise (Figs 6 and 7). Results of the cross-validations also indicate a slightly better performance of the statistical models based on the CIELAB data when compared to the models directly based on the RGB data (Table 1). These findings are consistent with multiple studies which suggest that the CIELAB colour space shows clear benefits for quantitative analyses (e.g. [43, 57]).

In the presented study we produced significantly better mapping results when training regression models for profiles EI and EVI individually. When applying the random forest regression analysis to the merged image data and sedimentological data of both profiles, much detail was removed from the resulting prediction maps (Fig 8). We assume that the high

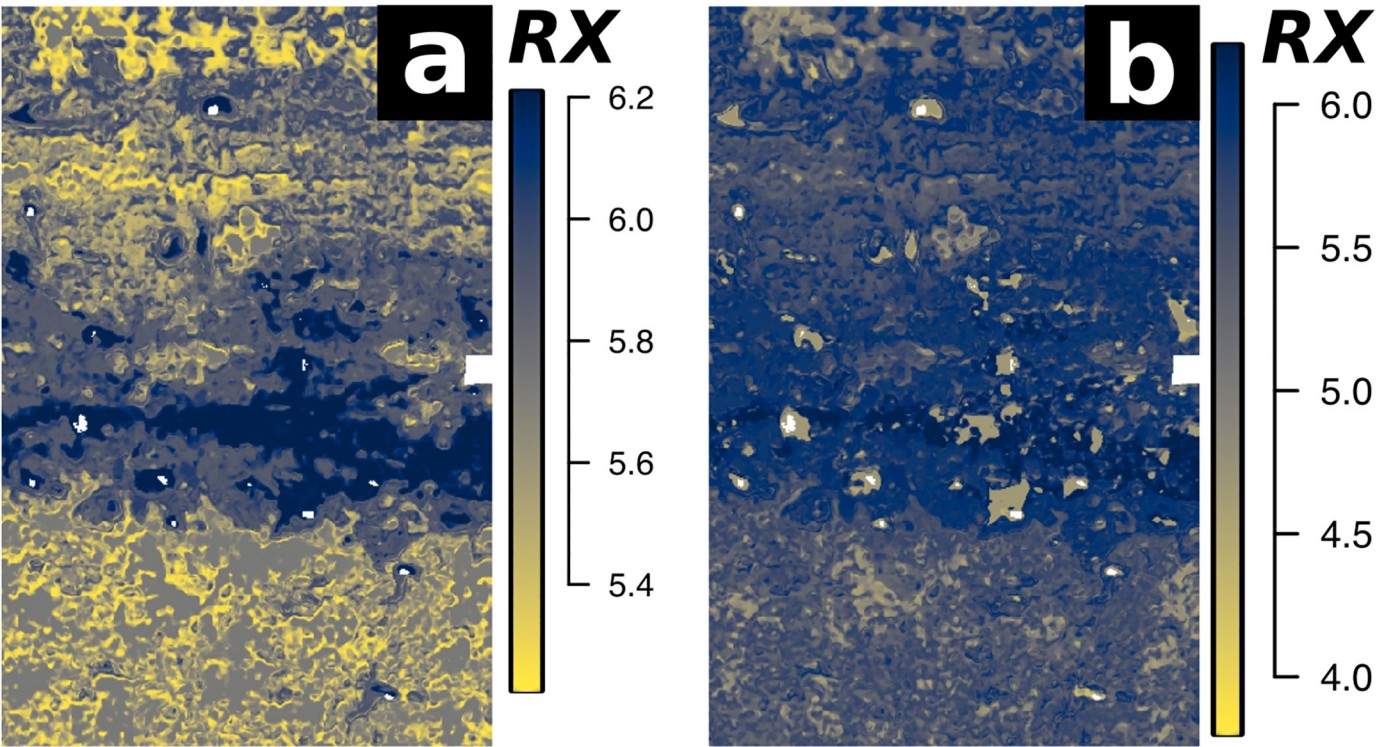

**Fig 8. Predicted weathering index values (RX = Ruxton Ratio) of Profile EI.** Prediction results from the regression models trained with (a) the hyperspectral image data of profile EI and (b) the combined hyperspectral image data of profiles EI and EVI. When analysing the profiles individually, significantly more detail is preserved (a).

contrast of profile EVI caused by just a few sediment properties resulted in these observed differences in modelling performance.

## Accuracy

Overall accuracy of the parameter maps derived from the statistical models was assessed (i) by visual comparison to the on-site stratigraphic interpretation (Fig 2), (ii) comparison of the mapping results to the laboratory results describing stratigraphic layers (Table 1, S1 Fig) and (iii) by performing leave-p-out cross-validation (CV), resulting in RMSE, MBE and pseudo $R^2$ values for each statistical model (Table 4). CV results of profile EI mostly agree with the quality of the mapping results of the parameters examined. They therefore help to quantify differences in the performance of the statistical models based on the image datasets used and likewise strengthen the significance of our results in general. Since our sampling strategy only allowed for the definition of highly local training and testing data, the CV results are influenced by spatial autocorrelation [63]. This becomes visible especially in profile EVI, where $R^2$ values are generally too high (Table 4). The selective character of the training data of the statistical models in combination with the distorted image data leads to an overfitting of the random forest model and thus produces the observed CV results. This is visible for multiple parameters in profile EVI: the high $R^2$ values for grain size, for example, disagree with the actual parameter maps, which are partly very inaccurate (S3 file). Another example is the disproportionally high $R^2$ value of 0.99 for the prediction of many parameters via the hyperspectral image data of profile EVI (Table 4). Possible solutions for this problem include tuning of the random forest model [64] or the usage of a spectral sensor that produces cleaner spectra and allows one to use a more specific or an extended feature space for training the statistical models used for prediction [e.g. 44]. Additionally our approach only included 0.56% (EI) and 0.28% (EVI) of the image pixels as training data. Increasing this number may lead to more reliable prediction results [65]. Taking into account the reliable results for profile EI and the acceptable results produced using the RGB or RGB derived CIELAB data for profile EVI, we suggest that the poorer performance of the proposed method for profile EVI originates in the distorted hyperspectral image data. For many of the parameters examined we were able to produce good prediction results comparable with those of past studies conducted under controlled conditions in the laboratory (e.g. [7]) or with a significantly higher number of samples (e.g. [12]). The expressiveness of the pseudo $R^2$ values used throughout the present study remains limited, but, in combination with the on-site layer delineation, the values led to a more traceable stratigraphic interpretation of the profiles examined.

## Conclusions

This study documents that the supervised regression of sediment parameters and—RGB and hyperspectral—image data provides valuable support for on-site sediment profile delineation and characterisation by an expert (here: archaeologist). The expert knowledge applied during on-site delineation of a sediment profile allows an integral analysis including the detailed description of certain macroscopic aspects such as texture, colour or compactness. Supervised regression involving image data is exclusively based on the physical spectral signal and requires the generation and application of several statistical models, mostly one per parameter contemplated. However, by applying these models, various chemical and textural sediment characteristics can be delineated across a profile and thus add information to the expert's delineation and interpretation of layers.

We demonstrate that the extrapolation of sediment properties of sampled material via image data can be transferred from the laboratory to archaeological fieldwork conditions. We

were able to create reliable maps of multiple sediment parameters (SOM, $Fe_2O_3$, $CaCO_3$, weathering, grain size) by applying statistical models for each parameter. The low number of samples furthermore renders our approach very limited in its destructiveness, which is crucial when researching anthropogenic heritage.

As the results imply, several characteristics of the proposed method must be considered: (i) the fieldwork situation needs to be carefully controlled, (ii) artificial light is highly recommended, and (iii) the spatial resolution of the image data and the sampled materials has to be adjusted according to the size, scale and complexity of the respective profile or planum examined. Bearing this in mind, our results suggest that spectral sensors in general, and particularly more advanced spectral sensors covering the NIR and MIR region in more detail [66], are suitable for the mapping of multiple sediment properties, commonly examined selectively during excavations. Our study illustrates the potential held by digital imaging as applied as a standard feature during most excavations nowadays. Treating digital images as physical measurements clearly offers a quick and low-cost way to increase the traceability of stratigraphic delineation and interpretations.

Future studies could also apply the proposed method to archaeological plana, possibly recorded by a multi- or hyperspectral sensor mounted on an unmanned aerial vehicle. However, as spectral information should serve to support profile delineation and not substitute it, we suggest a combined application of expert knowledge and spectral data.

## Supporting information

**S1 Fig. Sedimentological record of profiles EI (top) and EVI (bottom).** Water content of the sampled material was determined gravimetrically. Soil organic matter was calculated from quantitative measurements of total carbon (LECO TruspecCHN) and total inorganic carbon (Woesthoff Carmhograph C-16). Particle size distribution was measured using a laser diffraction particle size analyser (Beckmann-Coulter LS13 320). Element contents were measured using a portable energy-dispersive X-ray fluorescence spectrometer (Thermo Scientific Niton XL3t) and a inductively coupled plasma optical emission spectrometer (PerkinElmer Optima 2100 DV). Mineral composition was measured using an X-ray powder diffractometer (Rigaku MiniFlex 600). Weathering indices Ca-Ti-ratio (CTR), Ruxton-ratio (RX), Sesquioxide-ratio (SQ), K-Ti-ratio (KT) and Parker's Index (PI) were calculated. The methodology is described in more detail in the *Materials and methods* section.
(PNG)

**S1 File. Detailed sedimentological data along with plots showing measurement results from ICP-OES and p-ED-XRF as well as XRD plots.** Files available at: https://doi.org/10. 5281/zenodo.3906210.
(ZIP)

**S2 File. R-Scripts for the regression analyses performed throughout the presented study along with the data and the produced plots.** Files available at: https://doi.org/10.5281/ zenodo.3906216.
(TXT)

**S3 File.**
(TXT)

## Acknowledgments

We thank the Cluster of Excellence EXC264 Topoi (The Formation and Transformation of Space and Knowledge in Ancient Civilizations, Research Area A) for the interdisciplinary

framework which allowed us to conduct this research. Furthermore, we would like to thank the Sanaa Branch of the Orient Department of the DAI (Deutsches Archäologisches Institut) for their open-minded collaboration during fieldwork and the Geography Department of the Humboldt-Universität zu Berlin for the use of their spectroradiometer. We are very grateful to the Authority for Research and Conservation of Cultural Heritage (ARCCH) and the Tigray Culture and Tourism Bureau (TCTB) for permission to take samples and analyse them in Germany. Special thanks is expressed to Jan Krause, Björn Waske, Fabian Becker, Jacob Hardt, Moritz Nykamp, Henry Schubert, Robert Busch, Nadav Nir and Haftom Birhane for their support during fieldwork and the inspiring discussions. We would like to thank the reviewers for their thoughtful comments and efforts towards improving our manuscript.

## Author Contributions

**Conceptualization:** Vincent Haburaj, Brigitta Schütt.

**Data curation:** Vincent Haburaj, Sarah Japp, Philipp Hoelzmann, Brigitta Schütt.

**Formal analysis:** Vincent Haburaj.

**Funding acquisition:** Brigitta Schütt.

**Methodology:** Vincent Haburaj, Philipp Hoelzmann, Brigitta Schütt.

**Project administration:** Brigitta Schütt.

**Software:** Vincent Haburaj.

**Supervision:** Brigitta Schütt.

**Validation:** Vincent Haburaj, Sarah Japp, Iris Gerlach, Philipp Hoelzmann, Brigitta Schütt.

**Visualization:** Vincent Haburaj.

**Writing – original draft:** Vincent Haburaj.

**Writing – review & editing:** Vincent Haburaj, Sarah Japp, Iris Gerlach, Philipp Hoelzmann, Brigitta Schütt.

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
