## [Decision Letter · Decision Letter 0]

17 Jul 2020

PONE-D-20-19503

Coupling spectral imaging and laboratory analyses to digitally map sediment parameters and stratigraphic layers in Yeha, Ethiopia

PLOS ONE

Dear Dr. Haburaj,

Thank you for submitting your manuscript to PLOS ONE. After careful consideration, we feel that it has merit but does not fully meet PLOS ONE’s publication criteria as it currently stands. Therefore, we invite you to submit a revised version of the manuscript that addresses the points raised during the review process.

I agree with the reviewers' assertion that slight language editing remains needed. Both reviewers have made suggestions for this in their comments. Overall, this is a very solid, timely and well-presented contribution, only requiring very minor revisions. 

In addition to the reviewers' comments, I have one additional remark, which relates to your conclusion. In the final paragraph, you state that 'complementing the presented approach with unsupervised classification algorithms' would allow mapping stratigraphic layers semi-automatically. I think this statement needs further clarification (and for instance be moved to the discussion), or should be removed. It is unclear to me if you propose replacing supervised classification methods with unsupervised ones, or combining the current methodology with unsupervised approaches. If the former is the case, you do not present enough information in the paper to back this statement (particularly as you show under 'reproducibility' that the way in which training data are integrated (single profile/multi profile) affects the end result. How you would leap from this reduced accuracy to successful implementation of unsupervised approaches is unclear to me.

If, on the other hand, you propose a integrating unsupervised learning approaches into the current procedure, you should make explicit how you envision this precisely. In this case, and should you wish to elaborate on this issue, I suggest moving this part to the discussion. As this seems more a sidenote to your well presented methodology and results, it may be, in this case, perhaps be more suited to remove the statement.

Additionally, for Figure S1 (appendix 1) please provide information on the methodology to obtain the presented results in appendix 1. As this is separate from the text, and referred to under 'profile description' before the method section, this is a bit confusion. I suggest adding this information in the figure caption (where you can also refer to the method section).

We look forward to receiving your revised manuscript.

Kind regards,

Philippe De Smedt

Academic Editor

PLOS ONE

Journal Requirements:

2. In your manuscript, please provide additional information regarding the specimens used in your study. Ensure that you have reported specimen numbers and complete repository information, including museum name and geographic location.

For more information on PLOS ONE's requirements for paleontology and archaeology research, see https://journals.plos.org/plosone/s/submission-guidelines#loc-paleontology-and-archaeology-research.

In addition to the reviewers' comments, I have one additional remark, which relates to your conclusion. In the final paragraph, you state that 'complementing the presented approach with unsupervised classification algorithms' would allow mapping stratigraphic layers semi-automatically. I think this statement needs further clarification (and for instance be moved to the discussion), or should be removed. It is unclear to me if you propose replacing supervised classification methods with unsupervised ones, or combining the current methodology with unsupervised approaches. If the former is the case, you do not present enough information in the paper to back this statement (particularly as you show under 'reproducibility' that the way in which training data are integrated (single profile/multi profile) affects the end result. How you would leap from this reduced accuracy to successful implementation of unsupervised approaches is unclear to me.

If, on the other hand, you propose a integrating unsupervised learning approaches into the current procedure, you should make explicit how you envision this precisely. In this case, and should you wish to elaborate on this issue, I suggest moving this part to the discussion. As this seems more a sidenote to your well presented methodology and results, it may be, in this case, perhaps be more suited to remove the statement.

Additionally, for Figure S1 (appendix 1) please provide information on the methodology to obtain the presented results in appendix 1. As this is separate from the text, and referred to under 'profile description' before the method section, this is a bit confusion. I suggest adding this information in the figure caption (where you can also refer to the method section).

Reviewers' comments:

Reviewer's Responses to Questions

**Comments to the Author**

1. Is the manuscript technically sound, and do the data support the conclusions?

Reviewer #1: Yes

Reviewer #2: Yes

2. Has the statistical analysis been performed appropriately and rigorously? 

Reviewer #1: Yes

Reviewer #2: Yes

3. Have the authors made all data underlying the findings in their manuscript fully available?

Reviewer #1: Yes

Reviewer #2: Yes

4. Is the manuscript presented in an intelligible fashion and written in standard English?

Reviewer #1: Yes

Reviewer #2: Yes

5. Review Comments to the Author

Reviewer #1: This contribution presents a relevant study that was performed in a satisfactory manner. The manuscript is written in a transparent fashion, with its discussion and conclusions sufficiently supported by the data, argumentations, and literature. The descriptions of the materials and methods sections are exhaustive. The statistical analyses are suited to the types of data and performed satisfactorily. Inconsistent or unexpected results are discussed in the text. The figures are of high quality and appropriate in the light of the presented evidence. The data underlying the manuscript are available either within the manuscript or as supplementary information. The recommendations for future studies are supported by the rest of the manuscript.

There are a few minor grammatical and typographical errors and, while they do not interfere with the legibility of the manuscript, I would advise the authors to perform one additional round of revision, preferably by a native speaker or equivalent. In the Profile descriptions section, there is some inconsistency (in the main text) on the use of profile names, e.g. E I versus EI, etc. The same goes for instance for the capitalisation of hematite/Hematite. Regarding archaeological terminology, I recommend to replace the term 'cultural layers' with the less interpretation-heavy term 'anthropogenic deposits' or 'anthropogenic layers' (or possibly 'occupation deposits' if preferred), and the uses of the word 'cultural', for instance in conjunction with 'activity' or 'remains', by 'anthropogenic' as well in order to reflect more recent internationally accepted terminology.

Reviewer #2: This is an outstanding article that presents an innovative solution to mapping stratigraphy in archaeological sites using a robust quantitative approach based on imaging. I recommend it unreservedly for publication, other than some instances of awkward phrasing, which are particularly obvious in the early parts of the manuscript (and are highlighted in the attached marked up version of the manuscript).

6. PLOS authors have the option to publish the peer review history of their article (what does this mean?). If published, this will include your full peer review and any attached files.

Reviewer #1: No

Reviewer #2: **Yes: **Ian Moffat

---

## [Author Response · Author response to Decision Letter 0]

10 Aug 2020

Answer to Reviewer 1:

Thank you very much for reviewing our paper. We corrected the mentioned grammatical and typographical errors. The named passages were adjusted accordingly. Profile names are now consistent throughout the text. As suggested, the term ‘anthropogenic layers’ is now used instead of ‘cultural layers’. Furthermore, proof-reading was carried out by a native speaker and the text was changed in several passages to improve legibility and consistency.

Answer to Reviewer 2:

Thank you very much for reviewing our paper and highlighting several language errors. The mentioned passages were adjusted accordingly. Furthermore, proof-reading was carried out by a native speaker and the text was changed in several passages to improve legibility and consistency.

---

## [Editor Report · Decision Letter 1]

26 Aug 2020

Coupling spectral imaging and laboratory analyses to digitally map sediment parameters and stratigraphic layers in Yeha, Ethiopia

PONE-D-20-19503R1

Dear Dr. Haburaj,

We’re pleased to inform you that your manuscript has been judged scientifically suitable for publication and will be formally accepted for publication once it meets all outstanding technical requirements.

Kind regards,

Philippe De Smedt

Academic Editor

PLOS ONE
---

## [Editor Report · Acceptance letter]

27 Aug 2020

PONE-D-20-19503R1 

Coupling spectral imaging and laboratory analyses to digitally map sediment parameters and stratigraphic layers in Yeha, Ethiopia 

Dear Dr. Haburaj:

I'm pleased to inform you that your manuscript has been deemed suitable for publication in PLOS ONE. Congratulations! Your manuscript is now with our production department. 

Kind regards, 

on behalf of

Dr. Philippe De Smedt 

Academic Editor

PLOS ONE